# The Utility of 68Ga-PSMA PET/CT in Decisions Regarding Administering Salvage Radiotherapy to Men with Prostate Cancer

**DOI:** 10.3390/ijerph20010537

**Published:** 2022-12-29

**Authors:** Jennifer Ben Shimol, Ron Lewin, Zvi Symon, Barak Rosenzweig, Raya Leibowitz-Amit, Yael Eshet, Liran Domachevsky, Tima Davidson

**Affiliations:** 1Barzilai Medical Center, Ashqelon 7830604, Israel; 2Sackler Faculty of Medicine, Tel Aviv University, Tel Aviv 6997801, Israel; 3Department of Radiation Oncology, Chaim Sheba Medical Center, Tel Hashomer, Ramat Gan 5262000, Israel; 4Department of Urology, Chaim Sheba Medical Center, Tel Hashomer, Ramat Gan 5262000, Israel; 5Oncology Institute, Shamir Medical Center, Zerifin 7033001, Israel; 6Department of Nuclear Medicine, Chaim Sheba Medical Center, Tel Hashomer, Ramat Gan 5262000, Israel

**Keywords:** prostate cancer, salvage therapy, radiation therapy, 68Ga-PSMA, PET-CT

## Abstract

Background: Numerous papers have described 68Ga-prostate-specific membrane antigen (PSMA) positron emission tomography/computed tomography (PET/CT)’s sensitivity in identifying prostate cancer (PCa) recurrence. This study aimed to characterize the role of 68Ga-PSMA PET/CT in deciding to re-irradiate pelvic structures. Methods: 68Ga-PSMA PET/CT scans performed at Sheba Medical Center over seven years in 113 men were reviewed. All had undergone radiation to the prostate (70, 61.9%) or post-radical prostatectomy radiation to the prostate fossa (PF) (43, 48.1%), and had local or oligometastatic PCa recurrence and received salvage radiotherapy (SRT) based on PET/CT findings. Results: Mean age was 70.7 years. The mean grade group was 2.9; the mean prostate-specific antigen was 9.0. The 68Ga-PSMA PET/CT positive findings included: 37 (32.7%) in the prostate, 23 (20.4%) in seminal vesicles, 7 (6.2%) in the PF, and 3 (2.7%) in the seminal vesicle fossa. The mean standardized uptake value was 10.6 ± 10.2 (range: 1.4–61.6); the mean lesion size was 1.8 ± 3.5 mm (range: 0.5–5.1). SRT was directed toward the prostate and seminal vesicles in 48 (42.5%), PF in 18 (15.9%), and intrapelvic lymph node and bone in 47 (41.6%). Toxicities were mostly mild to moderate. Conclusion: 68Ga-PSMA PET/CT-identified relapse with targeted SRT was well-tolerated and may result in less onerous treatments.

## 1. Introduction

For men with localized prostate cancer, radical resection or radiation therapy (RT) serves as the primary form of treatment of localized prostate cancer [1]. Stereotactic body RT (SBRT), ultra-hypofractionated RT, has proven highly effective in curative, salvage, and adjuvant treatments of localized and oligometastatic prostate cancer; biochemical recurrence-free survival greater than 95% has been reported [2,3,4,5]. Nonetheless, SBRT carries a risk of invoking local harm, especially due to inconsistencies in patient positioning, target visualization, and target motion [6]. Moreover, more frequent acute genitourinary (GU) toxicity has been reported with SBRT than with conventional external beam RT (EBRT) [7]. Accordingly, clinicians have traditionally limited SBRT of the prostate to a single round.

The introduction of 68Ga-prostate-specific membrane antigen (PSMA) positron emission tomography/computed tomography (PET/CT) has enabled a more sensitive and specific appraisal of disease extent in men with prostate cancer [8,9]. To our awareness, however, no previously published report has specifically focused on the utility of 68Ga-PSMA in identifying sites of relapsed prostate cancer within the pelvis, in the setting of localized or oligometastatic disease, and determining suitable candidates for salvage RT (SRT). Accordingly, we aimed to describe the role of 68Ga-PSMA PET/CT in selecting the appropriate patients and identifying the pelvic sites most fitting for intrapelvic SRT, and in evaluating the subsequent toxicities.

## 2. Materials and Methods

### 2.1. Ethics

This single-institution study was approved by the institutional review board of Sheba Medical Center, according to the Declaration of Helsinki (approval no: SMC-20-7896). Informed consent was waived by the institutional review board of Sheba Medical Center due to the retrospective and anonymized nature of the study.

### 2.2. Study Design

The computerized database of Sheba Medical Center was searched for men with prostate cancer who had undergone pelvic SRT following the performance of PET/CT with the use of 68Ga-PSMA. The study period was 1 January 2014 through 22 March 2021. Once identified, relevant reports and images were individually reviewed to confirm and characterize 68Ga-PSMA-positive lesions within the pelvis. Imaging data were retrieved from the picture archive and communication system (PACS, Carestream Health 11.0, Rochester, NY, USA), and clinical data were obtained from the electronic medical records at our hospital. Clinical data, including medical history, laboratory work, and biopsy results were reviewed.

Study inclusion criteria were: (1) men aged 18 years and older; (2) a history of prostate cancer previously treated with local RT of the prostate or prostatic bed in men who had undergone radical prostatectomy; (3) the availability of a 68Ga-PSMA PET/CT scan performed following prostate RT, which identified evidence of local recurrence within the pelvis or oligometastatic disease (defined by up to five detected cancerous lesions); and (4) receipt of SRT targeted to intrapelvic structures. Exclusion criteria were: (1) the presence of abnormal findings in the pelvic area on 68Ga-PSMA PET/CT, which were deemed not suspicious of prostate cancer lesions; (2) evidence of widespread disease as defined by more than five sites of 68Ga-PSMA-positive uptake on PET/CT [10]; or (3) consensus of the tumor board that disease was overly advanced even when fewer lesions were present, thus precluding SRT.

### 2.3. PET/CT Image Acquisition

PET/CT scanning was performed using a combined PET-CT protocol with a 16-detector-row helical CT scanner (Gemini GXL, Phillips Healthcare, Eindhoven, The Netherlands). This scanner enables simultaneous acquisition of up to 45 transaxial PET images, with interslice spacing of 5 mm in one bed position; and provides an image from the vertex to the thigh in about 10 bed positions. The transaxial fields of view and pixel sizes of the PET images reconstructed for fusion were 57.6 cm and 4 mm, respectively, with a matrix size of 144 × 144 mm. The CT component was performed with oral and intravenous contrast media. The following technical parameters were used for CT imaging: pitch 0.8, gantry rotation speed 0.5, 120 kVp, 250 mAs, 3 mm slice thickness, and specific breath-holding instructions [11,12,13].

Patients received an intravenous injection of 148 MBq of 68Ga-PSMA. About 60 min later, CT images were obtained from the vertex to the mid-thigh for about 32 s. A contrast-enhanced CT scan was captured 60 s after injection of 2 mL/kg of non-ionic contrast material (Omnipaque 370 GE Healthcare, Eindhoven, The Netherlands). An emission PET scan followed in 3D acquisition mode for the same axial image range, 2.0–2.5 min per bed position. The diagnostic CT images were used for fusion with the PET data and to produce a map for attenuation correction. PET images were generated with CT attenuation correction utilizing a line of response protocol, and the reconstructed images were constructed for review (EWB, Extended Brilliance Workstation, Philips Medical Systems, Cleveland, OH, USA) [13].

### 2.4. Image Interpretation

All the available images were interpreted by experienced specialists in nuclear medicine and radiology and re-reviewed by one of the study co-authors with 20 years’ experience and dual certification in radiology and nuclear medicine. The readers were not blinded to clinical information. Consensus was achieved on the interpretation of all the images.

Findings were considered positive when imaging showed increased 68Ga-PSMA uptake that was not explained by the normal bio-distribution or uptake that was higher than the physiological uptake in the surrounding tissue. 68Ga-PSMA activity was quantified by calculating a maximum standardized uptake value (SUVmax). This was conducted by manually generating a region of interest over the sites of abnormally increased radioactive material activity. CT images were examined for abnormalities over the same areas of interest. Structures of interest included the prostate and post-resection prostate fossa (PF), seminal vesicles (SV), the post-surgical seminal vesicle fossa (SVF), pelvic bones, pelvic soft tissue, the bladder, the penis, and possible pelvic lymphadenopathy including the iliac (common, external, internal), mesorectal/presacral, retroperitoneal para-aortic, and inguinal chains [14].

### 2.5. Design of the Radiotherapy Protocol

For those who had undergone salvage RT following the failure of prostatectomy, the gross tumor volume (GTV) was defined as the 68Ga-PSMA-avid lesion in the prostatic fossa. The PF clinical target volume (CTV) was delineated according to the guidelines issued by the Radiation Therapy Oncology Group for post-prostatectomy RT. Planning target volume (PTV) was defined as CTV plus a 0.5 cm margin (PTV1) and GTV plus a 0.5 cm margin (PTV2). A hypo-fractionated regimen of 63 Gray (Gy) in 30 daily fractions (Fx) was given to the PTV1, with a simultaneous integrated boost of 69 Gy to the PTV2. Elective pelvic nodal irradiation was 51 Gy in 30 Fx.

Prostate salvage re-irradiation was delivered with SBRT. CTV was defined as the 68Ga-PSMA-avid lesion in the prostate (when evident), with a three to five mm margin for well-defined lateralized recurrences (focal salvage therapy); or as whole-gland treatment for large volume or bilateral recurrences. The PTV was defined as the CTV plus a 0.5 cm margin. Hydrogel spacers or endo-rectal balloons were utilized in all the men who underwent salvage re-irradiation to the prostate, and fiducial markers were placed in the prostate. The median dose to the prostate was 36.25 Gy in five every-other-day Fx. Regional recurrences following definitive RT or surgery were treated with a hypo-fractionated regimen of 50 Gy in 25 Fx to the relevant elective lymphatic chain, with a simultaneous integrated boost of 60 Gy to PET-positive nodes.

Spine and non-spine bone metastases were treated with SBRT. The GTV was defined as the 68Ga-PSMA-avid bony lesion. The CTV for vertebral lesions was delineated according to recommendations of the International Spine Radiosurgery Consortium (ISRC). For non-spine lesions, the CTV was a 2 cm expansion around the GTV, with no expansions into soft tissues and no additional margins for PTV. SBRT dose to bony metastases was 30 Gy in three every-other-day Fx.

Volumetric modulated arc therapy planning (RapidArc (RA), Varian medical systems. Palo Alto, CA, USA) was used. Pelvic RT was given with a full bladder and empty rectum. Image-guided RT using daily cone-beam CT was used.

### 2.6. Extraction of Demographic and Clinical Data

The data extracted from the medical records included age, cancer type, prostate specific antigen (PSA), Gleason score of the primary tumor lesion, systemic treatments, a history of prostate resection, and the length of follow-up. The scheme of RT was investigated including technique, dosage, and target.

### 2.7. Evaluation of Toxicities

Toxicity was reported using the Common Terminology Criteria for Adverse Events (CTCAE) version 4. Acute toxicity was defined as an adverse reaction occurring within three months of RT, and late toxicity as a reaction occurring three months or more after RT.

### 2.8. Statistical Analysis

The data were represented as means ± standard deviations for continuous variables and as percentages for categorical parameters. Student’s t-test was employed to analyze qualitative parameters. The chi-square test was used to evaluate quantitatively the significance of differences found between parameters. The analysis was performed using SPSS version 21.0 (SPSS, IBM, Armonk, NY, USA).

## 3. Results

### 3.1. Baseline Characteristics

#### 3.1.1. Study Participants

During the study period, 327 men with prostate cancer were referred to the radiation oncology department of our institution following RT of the prostate or PF, as part of surveillance or for suspicion of relapse; and subsequently underwent 68Ga-PSMA PET/CT surveillance. Of them, 214 (65.4%) were excluded from the analysis: 188 (87.9%) did not have 68Ga-PSMA positive findings within the pelvis that were subsequently irradiated and 26 (12.1%) had widespread disease based on the results of 68Ga-PSMA PET/CT.

#### 3.1.2. Disease Characteristics

The mean age of the 113 men who met study inclusion criteria was 70.7 ± 7.1 years (range: 52–85). Disease stage prior to the initial RT was available for 92 (81.4%) of the included men. Ten (11%) had stage one disease, 41 (45%) had stage two, 40 (44%) had stage three, and one man had stage four disease. The mean Gleason score for the cohort was 7.4 ± 1.1 (range: 5–10); the mean grade group was 2.9 ± 1.4. The mean PSA, available for 112 (99.1%), was 9.0 ± 20.6 (range: 0.02–170) (Table 1).

#### 3.1.3. Baseline Treatments

Forty-three (38.1%) men had previously undergone prostate resection. Sixty-one (54.0%) had received prior androgen deprivation therapy (ADT). The duration of ADT ranged from one week in a man for whom it was discontinued due to side effects, to 10 years. Three men who received ADT were also administered abiraterone acetate while one also received chemotherapy. In 38 (33.6%) men, initial RT was definitive; brachytherapy (BT) was utilized in two of them.

### 3.2. Initial Radiotherapy

#### Radiation Type and Dose

Among the 113 men included in this study, the type of initial RT was available for 106 (96.4%); 96 (93.2%) of the latter were treated with EBRT while 10 received BT. The dose directed toward the prostate or PF was available for all 98 men who had received EBRT. The mean dose administered to 97 of these men was 84.8 ± 29.1 Gy (range: 30.0–128.6); one man received 3000 Gy, which was not included in the calculation of mean. Of the 10 men who received BT, the dose was accessible for seven, for whom the mean dose was 159.3 ± 65.0 Gy (range: 125–305).

### 3.3. Response to the Initial Round of Radiation Therapy

#### 3.3.1. Biochemical Recurrence

The PSA value at the time of relapse was available for 103 (91.2%) men. The mean PSA was 3.7 ± 9.5 (range: 0.01–81.7). Fifty-seven men (55.2%) had evidence of biochemical failure, which prompted PET/CT evaluation. For the remaining 46 (44.7%) men, PET/CT was performed, as part of routine surveillance.

#### 3.3.2. 68Ga-PSMA PET/CT Findings at Recurrence

The sites of 68Ga-PSMA positive findings on PET/CT were available for all 113 men. Due to technical limitations, size and SUVmax were not available for two men. The mean SUVmax of the 163 detected lesions was 10.6 ± 10.2 (range: 1.4–61.6). The mean lesion size was 1.8 ± 3.5 cm (range: 0.5–5.1) (Table 2).

PET/CT illustrated 68Ga-PSMA avidity in the pelvic lymph nodes (LN) in 58 (51.3%) men, the prostate in 37 (32.7%), the SV in 23 (20.4%), the bony pelvis in 16 (14.2%), the PF in 7 (6.2%), and the SVF in 3 (2.7%) (Figure 1, Figure 2, Figure 3, Figure 4 and Figure 5). PET/CT also revealed abnormal uptake at the base of the penis in two men (Figure 6), the bladder base in one, the pelvic soft tissue in two, and along the omentum in one.

### 3.4. Salvage Radiation Therapy

#### 3.4.1. Type, Dosage, and Number of Rounds of Radiation

Of the 113 men included, 111 (98.2%) received salvage EBRT and two were administered salvage BT. For 31 (27.9%) of the former, RT was administered using SBRT. For 3 (2.7%) of the 111 men, SBRT was delivered using RA. The dose of radiation administered was available for 112 men. For 111 of them, the mean dose of RT was 60.4 ± 30.3 Gy (range: 16–186); one man received 5250 Gy and was therefore excluded from the calculation of the mean. The dosages of BT were unavailable.

Fourteen (12.4%) men underwent a subsequent round of SRT. The mean time between the first and second relapses was 25.9 ± 12.3 months. The two men who had received salvage BT were among those who received intrapelvic SBRT as part of another round of salvage treatment.

#### 3.4.2. The Site of Salvage Radiotherapy

SRT was directed toward the prostate and SV in 48 (42.5%) men; of whom RT was also administered to the LN in 9 (19%), the bony pelvis in 1, and the LN and pelvic bones in 1. SRT toward the PF was given to 18 (15.9%) men; three also received RT of the LN. SRT was delivered to the LN in 39 (34.5%) other men and to the pelvic bones in 2 (5.1%) of them. Eight (7.1%) were treated with SRT of the pelvic bones alone.

Among the nine men who received two rounds of SRT, initial SRT was administered to the PF in 4 (44%), the prostate and SV in 3 (33%), and the LN in 2 (22%). Subsequent SRT was directed toward the LN in 3 (33%), the prostate and SV in 2 (22%), the bony pelvis in 2 (22%), the PF in 1, and both the pelvic bones and LN in 1.

### 3.5. Follow-Up and Toxicities after Salvage Radiation

#### 3.5.1. Length of Follow-Up and Survival Rates

The mean follow-up time after SRT was 2.43 ± 2.0 years (range: 0.1–11). Six men were followed for three months or less and could therefore be evaluated only for acute toxic events. Ten (8.8%) men died over the follow-up period. Among those who died, the mean survival from the time of the detected relapse on 68Ga-PSMA PET/CT was 13 ± 28.6 months (range: 6–44).

#### 3.5.2. Acute Toxicities

Fifty-seven (50.4%) men experienced acute GU toxicities; 38 (67%) were grade 1 and 19 (33%) were grade 2, according to CTCAE version 4. Symptoms of frequency or urgency, and nocturia were most common, described by 41 (72%) of those with acute GU toxicity.

Acute gastrointestinal (GI) toxicities were relayed by 30 (26.5%) men; grade 1 in 28 (93.3%) and grade 2 (6.7%) in two. Loose stools were the most frequent side effects, occurring in 18 (60%) men. Other adverse events, including lethargy, pain, and anemia, were reported by 32 (28.3%) men; twenty-seven (84.4%) were grade 1 and five (15.6%) were grade 2.

#### 3.5.3. Late Toxicities

Forty-seven (43.9%) of the 107 men with sufficient follow-up time experienced late GU toxic events. Twenty-five (53.2%) were grade 2 and 21 (44.6%) grade 1; these were most frequently urinary frequency/urgency and nocturia. One man developed a grade 3 urinary stricture.

Ten (9%) men developed late GI adverse events; all were grade 1. Loose stools were most common, present in five (50%) men. Four (3.7%) other late grade 1 toxicities were reported, all complained of lethargy. Five (4.7%) additional grade 2 events were described; three men reported lethargy, one had lymphedema, and one developed a pelvic fracture following SRT of a bony metastasis.

## 4. Discussion

Prior studies have illustrated the superior sensitivity of 68Ga-PSMA over CT, MRI, and whole-body bone scans in staging and monitoring prostate cancer [15,16]. Our findings demonstrate the significance of 68Ga-PSMA PET/CT in identifying recurrence, selecting potential men for salvage RT, and defining target lesions with precision. Altogether, the safety profile of SRT was acceptable and offered an alternative to other treatments, often more invasive or hazardous.

The mean age of the men included in the study was 70 years, similar to prior articles that evaluated disease recurrence [17]. We found a broad range of Gleason scores, which is consistent with studies that reported an absence of correlation between Gleason score and the detection of relapse on 68Ga-PSMA PET/CT following RT [18]. For over 40% of our patients, relapse was revealed on 68Ga-PSMA PET/CT as part of routine surveillance in the absence of a rise in PSA. A mean PSA under four in our cohort corroborates reports that described high rates of relapse detection using68Ga-PSMA PET/CT in men with PSA scores below two [19,20]. Ninety percent of our patients had either stage two or three prostate cancer at presentation, slightly more had stage two. This concurs with another cohort of men with similar stages of cancer in which 68Ga-PSMA PET/CT demonstrated areas of disease [21]. More than half the men in that study had received ADT prior to baseline, though heterogeneity in duration and timing of treatment precluded reaching sound conclusions.

The mean SUVmax of lesions revealed with 68Ga-PSMA in this study parallels those reported by other groups [22]. Findings were most commonly identified in the pelvic LN, presenting in slightly over half our cohort. This is in agreement with other papers that examined the use of 68Ga-PSMA PET/CT in assessing recurrence, and highlights the particular benefit of detecting LN, even when lesions are sub-centimeter in size [23,24,25]. Lesions in the PF were identified in less than one fifth of the men who had undergone prostatectomy. This is similar to a previously reported rate of local failure in the PF [26]. Moreover, lymphadenopathy in combination with lesions in the PF were identified in close to 30% of our patients, mirroring reports that showed similarly high rates of recurrence in the pelvic LN following SRT of the PF [27].

In our cohort, the decision to re-irradiate focal targets was based on 68Ga-PSMA PET/CT findings. Likewise, previous research has shown that 68Ga-PSMA PET/CT-directed SRT is highly effective and safe over a follow-up of two years [28,29]. RapidArc was the most common form of initial RT employed, administered to approximately 40% of the men; this is consistent with reports of its safety and efficacy in the management of prostate cancer [30]. The mean dose of EBRT administered to the prostate or PF of 85% of our patients fell just short of 85 Gy, a dose previously shown to be effective [31].

Following SRT, acute grade 2 GU and GI toxicities occurred in 17% and 2% of the men, respectively. None of the acute events recorded were severe. Less than one-quarter of the men reported late grade 2 GU toxicities; one developed a severe urinary complication. All the late GI complaints that developed were mild in nature. This corroborates studies that demonstrated the safety of salvage SBRT, as evident by limited toxicities [32]. More than 90% of the included men survived the study period following disease. The rate of survival correlates with other papers that assessed oligo-recurrence using 68Ga-PSMA PET/CT [33].

This study has several limitations. Due to the retrospective design, the background data were occasionally incomplete, the follow-up time varied, and toxic events may have been recorded inconsistently. Moreover, the included population was heterogeneous, with differences in baseline treatments and initial RT protocols. A comparative population, in which only conventional imaging was used to guide clinical management was not examined. Finally, the mean follow-up time of 2.4 years did not enable a complete assessment of the long-term toxicities associated with SRT. To better evaluate the role of this imaging in decision-making and clinical outcomes, prospective studies should compare, among homogenous subgroups, the results of pelvic re-irradiation, with and without the use of 68Ga-PSMA PET/CT. Furthermore, data regarding toxic events should be collected over a longer duration, for a fuller evaluation of the safety of the described approach.

## 5. Conclusions

Notably, 68Ga-PSMA PET/CT plays a key role in demarcating sites of relapse and identifying men with the local or oligometastatic pelvic disease who are best suited for targeted SRT. Repeat RT proved safe with an acceptable toxicity profile. Our experience indicates that delineating a well-targeted treatment focus for re-irradiation results in an overall reduction in adverse events. The inclusion of 68Ga-PSMA PET/CT in the clinical management of men with prostate cancer offers guidance in the decision to re-irradiate and may reduce the need for other, more burdensome treatments.

## Figures and Tables

**Figure 1 ijerph-20-00537-f001:**
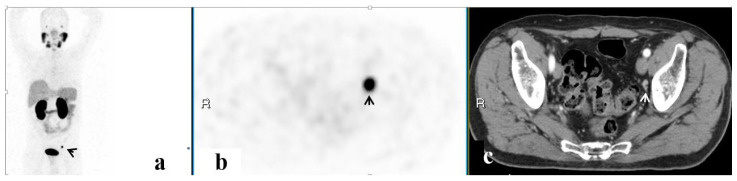
A PSMA-avid lesion in a solitary pelvic lymph node. 68Ga-PSMA scan, maximum intensity projection (MIP) (**a**) corresponding PET (**b**) and CT (**c**) axial slices demonstrating focally increased uptake in the solitary pelvic lymph node on the left (arrows).

**Figure 2 ijerph-20-00537-f002:**
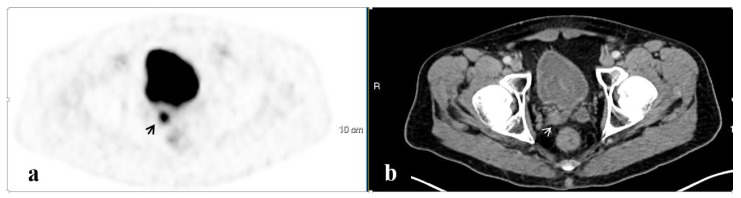
A PSMA-avid lesion in the seminal vesicles. 68Ga-PSMA scan, PET (**a**) and CT (**b**) axial slices demonstrating focally increased uptake in the seminal vesicles on the right (arrows).

**Figure 3 ijerph-20-00537-f003:**
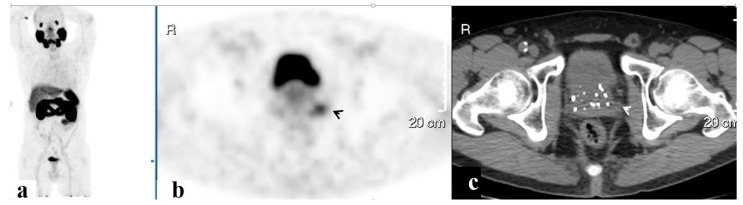
A PSMA-avid lesion in the prostate. 68Ga-PSMA scan, maximum intensity projection (MIP) (**a**) and corresponding PET (**b**) and CT (**c**) axial slices demonstrating focally increased uptake in the prostate on the left (arrows).

**Figure 4 ijerph-20-00537-f004:**
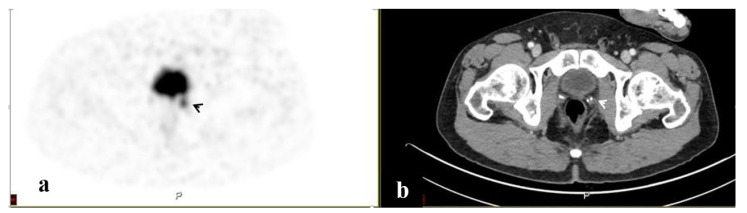
A PSMA-avid lesion in the pelvis at the prostatic bed. 68Ga-PSMA scan, PET (**a**) and CT (**b**) axial slices demonstrating focally increased uptake in the pelvis at the prostatic bed on the left (arrows).

**Figure 5 ijerph-20-00537-f005:**
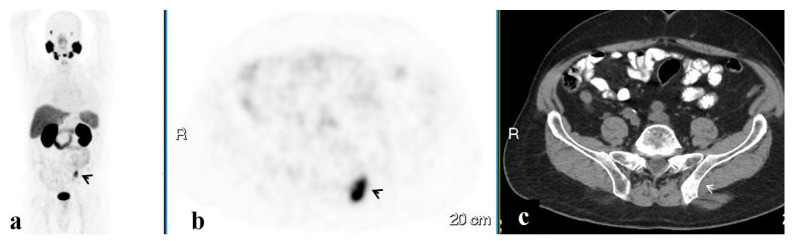
A PSMA-avid lesion in a blastic lesion in the iliac bone. 68Ga-PSMA scan, maximum intensity projection (MIP) (**a**) and corresponding PET (**b**) and CT (**c**) axial slices demonstrating focally increased uptake in the blastic lesion in the iliac bone on the left (arrows).

**Figure 6 ijerph-20-00537-f006:**
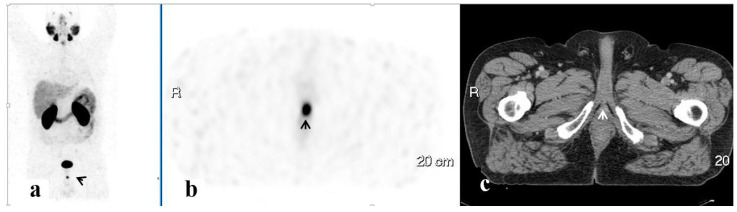
68Ga-PSMA scan, maximum intensity projection MIP (**a**) and corresponding PET (**b**) and CT (**c**) axial slices demonstrating focally increased uptake at the base of the penis (arrows).

**Table 1 ijerph-20-00537-t001:** Disease characteristics of the cohort prior to salvage radiotherapy.

**Mean Age at Diagnosis (y)**	70.7 ± 7.1				
**No. (%) who Underwent Prostatectomy**	43 (48.1%)				
**Mean Gleason Score**	7.4 ± 1.1				
**Grade Group**	1	2	3	4	5
**No. (%)**	27 (23.8)	25 (22.1)	18 (15.9)	19 (16.8)	24 (21.2)
**Initial RT Type ^1^**	RA	3D-CRT	IMRT	SBRT ^2^	BT
**No. (%)**	48 (45.3)	13 (12.2)	10 (9.4)	25 (23.6)	10 (9.4)
**Mean PSA at Relapse ^3^**	3.7 ± 9.5				

^1^ Available for 106 men; ^2^ Type unspecified; ^3^ Available for 103 men; BT: brachytherapy; IMRT: intensity modulated radiation therapy; No.: number; PSA: prostate specific antigen; RA: RapidArc; RT: radiotherapy; SBRT: stereotactic body radiation therapy; 3D-CRT: three-dimensional conformal radiation therapy; y: years.

**Table 2 ijerph-20-00537-t002:** 68Ga-PSMA PET/CT positive findings.

Site	No. of Men	Mean SUVmax ^1^	Mean Lesion Size (mm) ^1^	No. Accompanied by LN
Isolated pelvic LAD	43	11.0 ± 11.5	1.0 ± 0.3	n/a
Prostate	37	10.0 ± 6.4	2.1 ± 1.1	11
SV	23	8.6 ± 5.9	0.5 ± 0.5	7
Pelvic bones	16	14.2 ± 18.2	2.0 ± 1.0	2
PF	7	5.8 ± 5.6	1.4 ± 0.4	2
SVF	3	6.0 ± 1.6	1.3 ± 0.4	1
Other ^2^	6	8.0 ± 3.2	1.3 ± 0.4	2

^1^ Available for 111 men; ^2^ Penis, bladder, pelvic soft tissue, omentum; LAD: lymphadenopathy; n/a: non-applicable; LN: lymph node; No.: number; PF: prostate fossa; SUVmax: standardized uptake value; SV: seminal vesicles; SVF: seminal vesicle fossa.

## Data Availability

The data presented in this study are available on request form the corresponding author. The data are not publicly available dur to privacy restrictions.

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
