# Peer review of "The Utility of 68Ga-PSMA PET/CT in Decisions Regarding Administering Salvage Radiotherapy to Men with Prostate Cancer"

_ijerph, 2022, doi:10.3390/ijerph20010537_

Round 1

Reviewer 1 Report

The authors present their retrospective experience with PSMA-guided salvage re-radiation for recurrent prostate cancer. The study adds valuable information in this rapidly evolving clinical space. The manuscript is well written, but it might be improved by considering the following comments.

Abstract: Methods: The abstract does not make it clear that this study only includes patients who failed prior pelvic radiation and not patients who only underwent prostatectomy.

Abstract: Results: Sentence 2: Instead of reporting mean Gleason score, the authors should report mean grade group. Gleason score 7 could be grade group 2 or 3, and these have different prognoses and treatment options. This should also be reflected in the results section.

Methods: Why were men who had prior radical prostatectomy without radiation not included? This is the more common use of salvage radiation. Most men who have recurrence after radiation are considered for salvage prostatectomy or surgical excision of the recurrent lesions or lymph nodes, not additional radiation.

3.4.1: Sentence 2: Should this state “of the former,” instead of “of the latter”?

3.5.1: Sentence 3: What were the causes of death for these 10 men? Of the men who survived, how many of them were cancer free at the time of latest follow up?

Limitations: A mean follow-up of 2.4 years is inadequate to assess for long-term radiotherapy toxicitiy. This should be mentioned in the limitations section.

A mean follow-up of 2.4 years is also inadequate to assess for oncologic outcomes. Thus, there is no way of knowing if this PSMA-guided salvage radiation protocol is superior to standard of care. The authors should discuss this and the next research steps. How would you design a clinical trial to assess the true benefit of PSMA-guided salvage therapy?

Author Response

Thank you for your helpful suggestions. We implemented changes as described below.

The authors present their retrospective experience with PSMA-guided salvage re-radiation for recurrent prostate cancer. The study adds valuable information in this rapidly evolving clinical space. The manuscript is well written, but it might be improved by considering the following comments.

Abstract: Methods: The abstract does not make it clear that this study only includes patients who failed prior pelvic radiation and not patients who only underwent prostatectomy.

We agree with this comment and have rephrased the methods section of the abstract to more clearly describe the study population.

Abstract: Results: Sentence 2: Instead of reporting mean Gleason score, the authors should report mean grade group. Gleason score 7 could be grade group 2 or 3, and these have different prognoses and treatment options. This should also be reflected in the results section.

We are grateful for this suggestion. We have consequently replaced mean Gleason score with mean grade group in the results section of the abstract. Moreover, we have added the mean grade group to the results section and have elaborated on the grade group breakdown in table 1.

Methods: Why were men who had prior radical prostatectomy without radiation not included? This is the more common use of salvage radiation. Most men who have recurrence after radiation are considered for salvage prostatectomy or surgical excision of the recurrent lesions or lymph nodes, not additional radiation.

We appreciate the question and agree that evaluating the utility of PET/CT in men without prior pelvic radiation is fitting for continued study. However, the current study focused on PET/CT use in the scenario of re-irradiation, regardless of prior prostatectomy. This is due to concern regarding toxicities, specifically in the context of repeat pelvic irradiation. We believe that application of PET/CT to delineate pelvic sites for repeat irradiation, as described herein, reduces rates of adverse events. This point is emphasized in our conclusion.

3.4.1: Sentence 2: Should this state “of the former,” instead of “of the latter”?

Thank you for bringing this mistake to our attention. We corrected it accordingly.

3.5.1: Sentence 3: What were the causes of death for these 10 men? Of the men who survived, how many of them were cancer free at the time of latest follow up?

We recognize both questions to be of merit. However, we do not have access to the relevant data.

Limitations: A mean follow-up of 2.4 years is inadequate to assess for long-term radiotherapy toxicitiy. This should be mentioned in the limitations section.

We agree with the comment and included the short follow-up time in the limitations section.

A mean follow-up of 2.4 years is also inadequate to assess for oncologic outcomes. Thus, there is no way of knowing if this PSMA-guided salvage radiation protocol is superior to standard of care. The authors should discuss this and the next research steps. How would you design a clinical trial to assess the true benefit of PSMA-guided salvage therapy?

The suggestion to address future research steps is appreciated and we have done so accordingly.

Reviewer 2 Report

Thank you for inviting me to review the article titled “The utility of 68Ga-PSMA PET/CT in decisions regarding administering salvage radiotherapy to men with prostate cancer

The authors are reporting the use of PSMA PET scans to identify appropriate candidates for salvage RT at their institution and then reporting adverse event of salvage RT

Comments/Criticisms:

Major:  

There is no comparison to identify how much more utility PSMA PET adds compared to conventional imaging modality.

In order to identify the utility of PSMA PET scan (which is much more expensive than conventional imaging studies) for surveillance, there needs to be a clear comparison and demonstration of the superiority of the PSMA PET.

PSMA PET is somewhat commonly used at academic centers for patients with biochemical recurrence to identify patients to whom definitive localized therapy such as salvage RT can be offered.

Minor:

Methods mention that selection criteria ‘history of prostate cancer previously treated with local RT of the prostate’ but in Table-1, 43(48%) had a prostatectomy

Author Response

Thank you for your thoughtful comments. We have addressed them with the following revisions.

Comments/Criticisms:

Major:  

There is no comparison to identify how much more utility PSMA PET adds compared to conventional imaging modality.

In order to identify the utility of PSMA PET scan (which is much more expensive than conventional imaging studies) for surveillance, there needs to be a clear comparison and demonstration of the superiority of the PSMA PET.

PSMA PET is somewhat commonly used at academic centers for patients with biochemical recurrence to identify patients to whom definitive localized therapy such as salvage RT can be offered.

Thank you for your comment. We acknowledge the limitations of our study given the lack of comparison available and have spelled this out more clearly in the limitations section of the revised version. As you have correctly pointed out, our study was performed in an academic institution in which the use of PSMA PET is the standard of care for staging and monitoring prostate cancer, given its high sensitivity compared with conventional CT, and its ability to provide a full-body scan compared with MRI. Our retrospective study relies on the guidelines that emerged from Hofman MS et al. (https://pubmed.ncbi.nlm.nih.gov/32209449/), as the clinical question regarding the superiority of PSMA compared with conventional imaging was previously addressed. We added this source to our discussion.

Minor:

Methods mention that selection criteria ‘history of prostate cancer previously treated with local RT of the prostate’ but in Table-1, 43(48%) had a prostatectomy

We appreciate your noting this. A history of prostatectomy was not a criterion for inclusion or exclusion in our study. The men included were treated with radical prostatectomy initially, followed by radiation (and re-irradiation afterwards). To clarify this, we revised part of the methods of the abstract and elaborated the description of the inclusion criteria, to specify that prior history of prostatectomy with local radiation provided to the prostate bed rather than the prostate was permitted.

Round 2

Reviewer 2 Report

Thank you for your comments on the questions and for incorporating them in your paper.